# α_1_-Adrenergic Receptors: Insights into Potential Therapeutic Opportunities for COVID-19, Heart Failure, and Alzheimer’s Disease

**DOI:** 10.3390/ijms24044188

**Published:** 2023-02-20

**Authors:** Dianne M. Perez

**Affiliations:** The Lerner Research Institute, The Cleveland Clinic Foundation, 9500 Euclid Ave, Cleveland, OH 44195, USA; perezd@ccf.org; Tel.: +1-216-444-2058

**Keywords:** alpha adrenergic, GPCR, therapeutics, heart failure, Alzheimer’s, COVID-19

## Abstract

α_1_-Adrenergic receptors (ARs) are members of the G-Protein Coupled Receptor superfamily and with other related receptors (β and α_2_), they are involved in regulating the sympathetic nervous system through binding and activation by norepinephrine and epinephrine. Traditionally, α_1_-AR antagonists were first used as anti-hypertensives, as α_1_-AR activation increases vasoconstriction, but they are not a first-line use at present. The current usage of α_1_-AR antagonists increases urinary flow in benign prostatic hyperplasia. α_1_-AR agonists are used in septic shock, but the increased blood pressure response limits use for other conditions. However, with the advent of genetic-based animal models of the subtypes, drug design of highly selective ligands, scientists have discovered potentially newer uses for both agonists and antagonists of the α_1_-AR. In this review, we highlight newer treatment potential for α_1A_-AR agonists (heart failure, ischemia, and Alzheimer’s disease) and non-selective α_1_-AR antagonists (COVID-19/SARS, Parkinson’s disease, and posttraumatic stress disorder). While the studies reviewed here are still preclinical in cell lines and rodent disease models or have undergone initial clinical trials, potential therapeutics discussed here should not be used for non-approved conditions.

## 1. Introduction

Receptors that are activated by the adrenaline-type catecholamines, epinephrine (Epi) and norepinephrine (NE), are called adrenergic receptors (ARs). They belong to the G-Protein Coupled Receptor (GPCR) superfamily, which are receptors that transduce their intracellular signals through G-proteins. According to their physiological effects on the body, they were initially assigned as classifications α and β [1]. α-ARs were later further subdivided into α_1_- and α_2_-ARs, after noting that some functions were distinctively different between the two families. Upon further tissue characterization and molecular cloning, α_1_-ARs were further subdivided into the α_1A_-, α_1B_-AR, and α_1D_-AR subtypes based upon the subsequent cloning of the receptors [2,3,4]. The α_1C_-AR is missing from the current α_1_-AR nomenclature due to misclassification and incomplete pharmacological characterization of the α_1A_-AR subtype [4,5].

## 2. Pharmacology

α_1A_-ARs can be pharmacologically distinguished in tissues and cell lines from the α_1B_-AR subtype based upon a 10–100-fold higher binding affinity for several ligands that are commercially available [6,7] (Table 1). The α_1D_-ARs share more pharmacological similar and genetic homology with the α_1A_- than the α_1B_-AR but buspirone analogs (i.e., BMY7378) have been developed that have at least a 10-fold higher binding affinity for the α_1D_-AR over the α_1A_-AR subtype [8,9] and 100-fold selectivity compared with the α_1B_-AR subtype. α_1B_-AR does not have sufficiently selective ligands developed yet, but with the recent crystal structure of the α_1B_-AR bound with the antagonist cyclazosin [10], chiral analogs are being developed [11].

## 3. Signal Transduction

While the nine subtypes (α_1A_, α_1B_, α_1D_, α_2A_, α_2B_, α_2C_, α_1_, α_2_, and α_3_) bind Epi and NE with comparative affinities, the three different families couple to different G-proteins and effector pathways that allow specificity in function. While all GPCRs can couple to multiple G-proteins, they strongly couple to only a few. α-ARs couple more efficiently to Gα_s_, which stimulates adenylate cyclase and increases cAMP levels. α_2_-ARs are transduced by Gα_i_, which inhibits the production of cAMP. α_1_-ARs couple to Gαq to activate phospholipase C that causes the hydrolysis of membrane-bound phosphatidylinositol 4,5-bisphosphate to release inositol triphosphate (IP3) and diacylglycerol (DAG). IP3 binds to IP3 receptors located on the endoplasmic reticulum which causes the release of calcium. DAG activates protein kinase C (PKC), an enzymatic effector that can phosphorylate many proteins to amplify signals downstream in the signaling cascade. α_1_-ARs, as in all GPCRs, can signal directly or through cross-talk to couple to many other signaling pathways, both G-protein-dependent and independent, and through spatio-temporal as well as biased-agonistic mechanisms [13,14,15,16,17,18]

## 4. General Physiology

### Blood Pressure

The best described function of the α_1_-AR activation is to increase blood pressure via the contraction of the vasculature which highly expresses α_1_-ARs in the smooth muscle layer [19]. α_1_-ARs regulate blood pressure through IP-mediated increased calcium release, causing the contraction of the vascular smooth muscle by activating myosin light chain kinase and actin/myosin cross-bridge formation [20], and may involve several different signaling pathways involving PKC, PI3K, Rho Kinase, and MAPK [21,22]. Transgenic and KO mice have been developed for all three α_1_-AR subtypes, using receptors that are WT or contain constitutively-active mutations, some have cardiac-specific promoters and others that are systemically or conditionally expressed [23,24,25,26,27,28,29,30,31,32,33,34,35,36,37,38]. These mouse models provided various insights into the physiological differences between the subtypes. Using these mouse models, all of the α_1_-AR subtypes have been reported to affect phenylephrine-induced blood pressure [24,33,39] but only the α_1D_-AR KO decreased resting blood pressure [39,40].

## 5. α_1A_-AR Agonists

### 5.1. Currently Approved Uses

α_1_-AR agonists are not commonly prescribed because of the potential to raise blood pressure but are approved for the treatment of vasodilatory shock, hypotension, hypoperfusion, septic and refractory shock, and cardiopulmonary arrest. Approximately 7% of critically ill patients develop refractory shock causing a 50% short-term mortality rate [41]. Vasopressor agents used to maintain blood pressure and preserve tissue perfusion during shock are methoxamine (discontinued in the US) or norepinephrine/epinephrine [42,43]. α_1_-AR agonists such as phenylephrine have been used in procedures to dilate the iris [44]. Phenylephrine, naphazoline, and oxymetazoline are also used in nasal decongestion and edema [45,46] and the facial erythema associated with rosacea [47,48].

### 5.2. Heart Failure and Cardioprotection

The human heart contains both the α_1A_ and α_1B_-AR subtypes with a total density of approximately 11–60 fmoles [49,50,51]. The α_1D_-AR may be present in the myocyte but at very low levels [52,53]. The current hypothesis is that selective α_1A_-AR agonists may be a potential treatment in heart failure [54,55], since chronic α_1B_-AR stimulation, as evidenced through transgenic mouse models, appears to be maladaptive by inducing dilated cardiomyopathy [29] or heart failure [37]. While α-AR blockers are a current treatment option for heart failure, using α_1A_-AR selective agonists may provide potentially greater benefits such as preventing dementia [56], improving metabolic function and glucose tolerance [56,57,58], increasing lifespan with reduce cancer risk [59,60] and reducing inflammation and cataracts [58,61].

The preclinical evidence that the α_1A_-AR subtype is cardioprotective and could be therapeutic for heart failure is abundant. Transgenic mice with heart-targeted α_1A_-AR overexpression were protected from dysfunction due to myocardial infarction [26], pressure-overload [25], or imparted ischemic preconditioning [34,62]. Correspondingly, α_1A_-AR KO mice had induced greater heart injury after myocardial infarction [55]. The α_1A_-AR selective agonists, A61603 or dabuzalgron, prevented damage from the cardiotoxic agent, doxorubicin [63,64,65] and increased contraction during heart failure [66]. Removing load by mechanical assist devices in failing human hearts improved function and re-distributed α_1A_-ARs from the peri- to intra-myocyte location [67]. However, there are currently no clinical trials underway, most likely due to the potential to increase blood pressure and the risk of stroke. The use of positive allosteric modulators (PAMs) for the α_1A_-AR developed to treat Alzheimer’s disease [12] are currently in preclinical studies in mice and to assess potential benefits in heart failure.

The ability of the α_1A_- and not the α_1B_-AR to cardioprotect may be due to several mechanisms. One is the ability of the α_1A_-AR to increase inotropy [30,68,69]. Another mechanism may be due to increased glucose uptake and oxidation in the heart [70] as glucose oxidation has been shown to repair heart damage after ischemia or heart failure [71,72,73,74,75,76]. Transgenic α_1A_- but not α_1B_-AR mice increased glucose uptake into the heart and only the α_1A_-AR KO mice displayed decreased glucose uptake into the heart [57]. Heart failure has been described as a metabolic disease of energy starvation [77] and so any therapeutic that can increase ATP production may improve heart function.

### 5.3. Cognition and Memory

α_1_-ARs have long been associated with learning and memory functions [7]. α_1_-AR agonists promoted while α_1_-AR antagonists blocked long-term potentiation (LTP, a mechanism of memory formation) in the rat CA1 hippocampus [78], neocortex [79], and may coordinate with β-AR signaling [80,81,82,83]. α_1A_-AR systemically overexpressing transgenic mice increased synaptic plasticity, LTP, and performance in a battery of cognitive tests of spatial memory, while α_1A_-AR KO mice performed poorly [60]. α_1B_-AR KO mice had impaired spatial learning to novelty and exploration [84], and a decrease in memory consolidation and fear-motivated exploration [85]. While α_1D_-AR KO mice did not show deficits in spatial learning [86], they did show deficits in working memory and attention [87]. While all three α_1_-AR subtypes are localized in the brain and expressed in overlapping domains, the α_1A_-AR subtype appears to have greater expression in cognitive areas such as the hippocampus and amygdala, as well as particular areas of the cortex and neurogenic regions involved in learning and memory [88,89]. The α_1A_-AR selective agonist cirazoline increased cognition and BrdU incorporation in normal adult mice, while the α_1A_-AR overexpressing transgenic mice had increased BrdU incorporation in both the subventricular and subgranular neurogenic regions [88].

In order to develop suitable therapeutic α_1A_-AR agonists to treat heart failure, cardiac ischemia, or Alzheimer’s disease, PAMs with sufficient signal bias would need to be developed that could regulate heart or brain function without effects on the vascular system to increase blood pressure. PAMs will increase a receptor activation and function but in such a way that it does not bind to the same site as the endogenous agonist (i.e., orthosteric), such as NE [90]. Allosteric modulators result in decreased side effects and have greater selectivity by binding to non-conserved regions of the receptor resulting in conformational bias that can alter the receptor’s signaling pathways. There are now many GPCR allosteric modulators in clinical trials [91]. Another issue is the poor brain penetration of most of the current α_1_-AR agonists which limit their use in neurological conditions. The first PAM at the α_1_-ARs with high selectivity for the α_1A_-AR subtype has been developed [12] that can cross the blood–brain barrier sufficiently enough to improve cognitive functions and modify disease in Alzheimer’s disease mouse models without increased blood pressure. This drug (i.e., Cmpd-3, Table 1) only activates the NE-bound receptor and can potentiate cAMP signaling without effects on IP-signaling. IP-signaling and the resulting calcium release causes the increase in blood pressure. However, NE-mediated cAMP signaling in the brain regulates learning and memory [92,93,94,95,96,97]. This drug is currently in preclinical studies to treat heart failure.

## 6. α_1_-AR Antagonists

### 6.1. Currently Approved Uses

As in the vascular system, α_1_-AR antagonists affect the contraction of smooth muscle in several organ systems. α_1_-AR blockage results in the relaxation of smooth muscle in the prostate and ureter to increase urinary flow [98,99,100]. Since the 1980s and 1990s, α_1_-AR antagonists are frequently used medications in the management of benign prostatic hyperplasia (BPH), kidney stones, and in therapy-resistant arterial hypertension, two conditions frequently found in older adults. As a powerful anti-hypertensive, α_1_-AR antagonists are not recommended as a first-line treatment [101,102] as they are counter indicative for those with heart disease. While α_1_-AR antagonists are effective in the relief of urinary symptoms and improve the quality of life in BPH, they appear less effective in preventing disease progression [103,104]. α_1_-AR blockers are also used to treat pheochromocytoma, a rare condition where a tumor forms on the adrenal gland or other paraganglia to cause excessive catecholamine release and severe hypertension. The tumor is excised immediately under the use of an α_1_-AR blocker to reduce hemodynamic instability, morbidity and mortality [105]. General counterindications for α_1_-AR antagonists will be discussed at the end of this article.

### 6.2. COVID-19/SARS

Coronavirus disease 2019 (COVID-19) and the causative agent, severe acute respiratory syndrome coronavirus 2 (SARS), can elicit a vigorous systemic immune response (i.e., hyperinflammation) in the lungs as well as multiple organs, resulting in heart and kidney failure, liver damage, precipitating severe illness, and increased mortality [106]. Recent evidence suggests that some patients with COVID-19 develop a cytokine storm syndrome that is associated with increased release of pro-inflammatory cytokines, disease severity, and poor clinical outcomes [107].

Beyond their role in neurotransmission, cardiovascular, and the stress response, α_1_-ARs have been shown to modulate the immune system [108,109], innate immunity [110], and inflammatory damage by increasing cytokine production in immune cells [111,112]. α_1_-ARs have been identified on a wide variety of immune cells. Identification of immune cells using flow cytometry depends upon highly avid antibodies whose specificity are questioned for the current commercially available antibodies for the α_1_-ARs and many other GPCRs [113]. However, many studies have utilized mRNA expression and ligand binding analysis. Human neutrophils contain the mRNA for all three α_1_-AR subtypes [114]. Monocytes contain the mRNA for the α_1B_- and α_1D_-ARs [112,115,116]. NK killer cells, leukocytes [117,118,119], and lymphocytes, including human peripheral blood lymphocytes [120,121], also contain α_1_-ARs but the subtypes are not clearly defined.

#### 6.2.1. α_1_-AR Antagonists May Protect against Severe COVID-19

Several studies indicate that α_1_-AR antagonists may reduce morbidity and mortality in patients at risk for hyperinflammation and cytokine storm that is often associated with COVID-19 and other conditions that result in severe respiratory tract conditions. Blockade of α_1_-AR function with prazosin prevents cytokine storm following pro-inflammatory conditions and increases survival in preclinical studies [122]. A retrospective analysis in two large cohorts of patients with acute respiratory distress (*n* = 18,547) and three cohorts with pneumonia (*n* = 400,907) found that patients exposed to α_1_-AR antagonists had a significantly lower risk (34%) for mechanical ventilation and death [123]. Similar results were obtained in a subsequent retrospective analysis on US veterans [124] and another large cohort study of influenza or pneumonia patients in Denmark [125]. These studies led to a clinical trial to test whether prazosin can prevent the cytokine storm syndrome [126] caused by COVID-19 (https://clinicaltrials.gov/ct2/show/NCT04365257, accessed on 7 February 2023) but this trial is currently halted due to lack of recruitment. These results extend circumstantial findings that prazosin may be an early preemptive therapy in COVID-19 and may prevent the cytokine storm and severe complications due to hyperinflammation.

#### 6.2.2. α_1_-AR Antagonists May Not Prevent COVID-19 Infection

The protective effects of α_1_-AR blockers against COVID-19 were recently challenged in a study using meta-analysis of millions of patients prescribed α_1_-AR blockers (alfuzosin, doxazosin, prazosin, silodosin, tamsulosin, and terazosin), compared to alternative medications (dutasteride, finasteride, and 5-α-reductase inhibitors) or tadalafil (PDE5 inhibitor) to treat BPH. This study found no reduction in the risk of COVID-19 infection due to the sustained use of α_1_-AR blockers [127]. The negative results are unlikely due to the comparison to non-α_1_-AR blocker treatments for BPH as the study of Thomsen et al. [125] also included non-users (normal controls). However, this study did find significant but not large differences on the ability of α_1_-AR blockers to confer protective benefits against death and ICU admission due to COVID-19.

The study of Nishimura et al. [127] suggested that previous positive results from clinical trials had systematic biases from residual confounding [128,129]. For example, patients with severe asthma are more likely to be prescribed α-agonists and to die from their asthma than patients with less severe disease but not receiving treatment. Therefore, such confounding would make α-agonists appear they were associated with asthma mortality. However, all epidemiology studies that utilize user vs. non-user comparisons from databases are prone to systematic biases from residual confounding. The study of Nishimura et al. [127] used a database of older male patients that are at higher-risk for COVID-19 and for developing severe COVID-19 compared to the general population, and then analyzed the risks of developing COVID-19, being hospitalized, or hospitalizations that also require intensive services requiring ventilation or oxygenation. The study of Thomsen et al. [125] and others, while also analyzing older men, used a database of high-risk patients already hospitalized with hyperinflammation or cytokine storm (pneumonia, severe COVID, and influenza) and measured α_1_-AR blocker effects on more severe outcomes (ICU, mortality). Therefore, one interpretation is that α_1_-AR blockers do confer protection, but the amount of pre-emptive protection is not that significant for use in the general population but only for a subset of severely ill patients once the cytokine storm has developed, and then used to reduce mortality. All of these studies have limitations in that they measured outcomes on men who are more likely to be prescribed α_1_-AR blockers due to BPH and may not reflect possible outcomes for women. Nevertheless, these results suggest the need for further clinical trials to include women and whether α_1_-AR blockers first ameliorates the severe symptoms of lower respiratory tract infection-associated hyperinflammation and the risk of death.

#### 6.2.3. The Case for Anti-Hyperinflammation as a Direct α_1_-AR Mediated Effect

There is precedent in preclinical studies for the ability of α_1_-AR blockers to reduce hyperinflammation. Prazosin prevents cerebral infarction by inhibition of the inflammatory cascade [130]. One mechanism that α_1_-ARs may use to combat hyperinflammation is through their association with chemokine receptors. Chemokines are a group within the cytokine family whose general function is to induce cell migration and are potential therapeutic targets in numerous inflammatory diseases, such as COVID-19. Several chemokine genes have been associated with disease severity and susceptibility to infection with COVID-19 [131]. At least 20 members of the human chemokine receptor family heterodimerize with the α_1B_ or α_1D_-AR subtypes and inhibited their function and were detectable in human monocytes [118]. The CXCR2 has been reported to heterodimerize with the α_1A_-AR in prostatic smooth muscle [132]. Many GPCRs can form homo- and hetero-oligomers, which is thought to alter their pharmacological behavior and function and may play a role in pathophysiology [133,134,135]. Another mechanism that is described is through catecholamine excess [136]. In animal studies, the blockade of catecholamine synthesis (and indirect blockage of α_1_-ARs) reduced cytokine release and protected mice against COVID-19 lethal complications [122]. Furthermore, autoantibodies against GPCRs, including the α_1_-AR, were observed in patients after SARS infection and suggested to cause impaired blood flow, the formation of microclots, and autoimmune dysfunction contributing to long-COVID symptoms [137,138]. These results suggest a direct effect of α_1_-AR antagonists in blocking α_1_-AR mediated adverse effects in hyperinflammation.

#### 6.2.4. The Case for Non-α_1_-AR Mediated Effects of Quinazoline Antagonists: PGK1

It is possible that the protective anti-inflammatory effects of prazosin, doxazosin and terazosin may be non-α_1_-AR mediated through activation of phosphoglycerate kinase (PGK1)-mediated ATP production. Terazosin and its related “osins” are postulated to mediate protective mechanisms by binding adjacent to the ADP-ATP site of PKG1 and facilitating its activation. PGK1 is the first enzyme in glycolysis where ADP enters the cleft of the active site and is converted into ATP and shown to inhibit apoptosis [139,140]. Terazosin increases the release of ATP by competing for the same binding site, re-exposing the binding pocket, thereby exerting an agonistic effect [140]. PKG1 binding and activation has also been demonstrated in related α_1_-AR antagonists that contain quinazoline motifs, such as alfuzosin, prazosin, and doxazosin [140]. PGK1 activation may improve cellular functions in disorders with an established energy deficit, common with critically ill patients [141] and COVID-19 patients [142,143]. Terazosin was shown to increase PGK1 activity and glycolysis in motor neuron models of amyotrophic lateral sclerosis (ALS), which correlated with protection and survival [144]. The effects of prazosin-like compounds appear directed at the quinazoline structural motif, as tamulosin, also an α_1_-AR blocker but with some selectivity for the α_1A_-AR [145], does not appear to mediate anti-inflammatory effects, does not contain the quinazoline motif, and does not interact with PGK1 [139,146]. Furthermore, an analysis of the Truven database and Danish nationwide health registries demonstrated that individuals treated with terazosin, alfuzosin, or doxazosin showed lower rates of Parkinson’s disease (PD) and PD-related diagnoses when compared with patients treated with tamsulosin [147]. Therefore, quinazoline-based antagonists of the α_1_-ARs may confer therapeutic levels of protection against inflammation and morbidity through non-α_1_-AR -mediated effects of increasing glucose metabolism by binding to the active site of PGK1.

While the above protective effects of PGK1 appear to be metabolic, α_1_-AR quinazolines (i.e., not tamsulosin) have also been shown in several studies to induce apoptosis in different cell lines and in vivo through non-α_1_-AR mechanisms [148,149,150]. Pyroptosis, a proinflammatory form of apoptosis, acts as a host defense mechanism against infections. Pyroptosis decreases the replicative ability of viruses by inducing the apoptosis of infected cells and exposing the virus to extracellular immune defenses. Several therapeutics that target inflammasomes, caspases, or cytokines are in clinical trials to evaluate efficacy in mitigating the severe outcomes of COVID-19 [151]. Therefore, the ability to reduce severity of COVID-19 outcomes by prazosin and other quinazolines may be due to their ability to increase apoptosis, improve energy deficit, or both.

These two different but protective mechanisms (metabolic verses apoptotic) may be cell-type, α_1_-AR subtype, or disease-dependent. All of the pro-apoptotic effects of quinazolines are non-α_1_-AR mediated and mostly found in cancer cell lines, while metabolic effects are more systematic and may be α_1_-AR subtype dependent. The non-quinazoline tamsulosin does not exhibit cytotoxic or apoptotic activity in cancer cell lines [148]. Prazosin treatment protects the brain by decreasing oxidative stress and apoptotic pathways [152]. A non-quinazoline α_1_-AR antagonist reduced inflammation and immune cell infiltration and improved insulin signaling in the adipose of fructose-fed rats [153], as well as cardiac, vascular, and renal dysfunction in hypertensive rats [154].

### 6.3. α_1A_-AR Activation but α_1B_-AR Blockage Is Protective

Concerning α_1_-AR subtype-dependent effects of antagonists, there is evidence that α_1A_-AR activation is protective, while chronic α_1B_-AR activation is damaging and neurodegenerative. Therefore, α_1A_-AR agonists would be protective and in systems where chronic α_1B_-AR activation is damaging, non-selective blockers may exert protective effects. Systemic overexpression of the α_1A_-AR in mice has anti-tumor effects [59], preconditions the heart against ischemia [34], reverses heart failure and cardiac apoptosis [62,65,66], and increases longevity [59]. In contrast, systemic overexpression of the α_1B_-AR subtype in mice was neurogenerative, induced autonomic dysfunction, heart failure, apoptosis [37,38,155,156], and decreased lifespan [59]. Tamsulosin has a 10-fold higher binding affinity and slower dissociation kinetics compared to the other two subtypes, rendering it an α_1A_-AR selective antagonist [145,157]. The epidemiology study of [158], while finding that usage of terazosin/alfuzosin/doxazosin failed to see any changes in the risk in Parkinson’s disease (PD) development, did find that tamulosin increased PD risk and may associate with disease progression. Protective effects of prazosin may be due to α_1B_-AR blockage since tamsulosin (α_1A_-AR blockage) does not induce apoptosis nor binds with PGK1. The study of Koenecke (2021) [123] found that doxazosin was two-fold more efficacious than tamsulosin in preventing COVID mortality, suggesting blockage of α_1B_ or α_1D_-mediated pro-inflammatory effects. There is an increased expression and cellular proliferation of the α_1B_-AR subtype in prostatic cancer cell lines that exhibit apoptosis with prazosin [159]. α_1B_-AR activation mediates unchecked cell cycle progression and induced foci formation [160], supporting a cancer-inducing paradigm. Therefore, protective effects of α_1_-AR blockage might indicate that the α_1B_- or α_1D_-AR subtype is being blocked in the particular tissue or disease.

### 6.4. Other Neurological Benefits of α_1_-AR Quinazoline Antagonists: Parkinson’s, ALS, PTSD

Neuroprotection, just like cardioprotection, may be mediated through increased metabolism [161]. As the heart is energy-starved during failure, so too are several neurodegenerative diseases. Glucose metabolism is essential for proper brain function, accounting for 20% of whole-body energy consumption, but compiles only 2% of body mass. Therefore, brain energy demand is mostly met by the metabolism of glucose [162]. Bioenergetic and mitochondrial dysfunction are common hallmarks of PD and ALS, and regulate disease onset and progression [161,163,164]. In ALS pathogenesis, the early dysregulation of the AMPK signaling pathway was found in motor neurons and in a large proportion of patients [165]. Preclinical and epidemiologic data suggest that terazosin, a quinazoline antagonist, may be neuroprotective in PD and ALS [144,166] and impart a decreased risk for developing PD [139]. However, another study that analyzed a large database of terazosin/alfuzosin/doxazosin users failed to see any changes in the risk of PD development [158]. A clinical study evaluating the safety and tolerability of terazosin, 5 mg once daily for 12 weeks, in patients with PD has been initiated (NCT03905811). Doxazosin can also reduce oxidative stress, pro-inflammatory cytokines, and cell death in rat photoreceptor cells in vivo [167]. Terazosin protected against organ damage, sepsis, and death in rodent models [140]. Therefore, nonselective α_1_-AR quinazoline antagonists may also be useful in other neurodegenerative diseases.

Posttraumatic stress disorder (PTSD) is associated with elevated noradrenergic activity [168,169,170]. In clinical trials and meta-analysis, prazosin has been effective and well-tolerated to reduce combat trauma nightmares, sleep disorders, and general clinical status in veterans [171,172,173] and for general trauma-related nightmares [174]. Compared with image rehearsal therapy which is the recommended treatment for trauma-induced nightmares, prazosin was more efficacious at relieving the frequency and stress-related symptoms but image rehearsal therapy combined with cognitive behavioral therapy was better at improving sleep quality [175]. A more recent study by Raskind et al. (2018) [176] also showed that prazosin did not improve sleep-related problems in PTSD. However, it is unclear whether or not prazosin will reduce the risk of nightmares in people without trauma or whether other α_1_-AR blockers (non-quinazolines) are effective. α_1A_-AR stimulation has been suggested to mediate stress-induced memory formation and consolidation [7] and, therefore, blockage with prazosin may be psychotherapeutic, resulting from a direct α_1A_-AR antagonistic effect.

## 7. Counterindications

### 7.1. α_1A_-AR Blockers but Not Non-Selective Antagonists May Increase Dementia and Depression

While non-selective α_1_-AR quinazoline antagonists appear to improve symptoms in neurodegenerative diseases and PTSD, regardless of whether they are α_1_-AR or non-α_1_-AR mediated, antagonists that are selective for the α_1A_-AR subtype may potentiate neurodegeneration and dementia. This would be consistent with α_1A_-AR activation demonstrating increased cognitive performance and reversing Alzheimer’s disease as discussed in this review. Just as tamsulosin does not follow the protective properties of quinazoline antagonists as discussed in the above sections, tamsulosin, which is α_1A_-AR selective, increases the risk of dementia modestly and other adverse cognitive effects, in particular among patients over age 61 [177]. This study utilized cohorts taking various medications (including 5a-reductase and quinazoline α_1_-AR blockers) for BPH as well as those taking no medications and followed them for 20 months after the first prescription was filled. However, two subsequent clinical studies contradict these results [178,179]. While tamsulosin did increase the risk of dementia, there was no evidence of a dose–response, and after adjustments for confounding variables, the results were not significant [179]. Differences between the three studies could be due to the mean age that was assessed. The two negative studies used a mean age of 78.7 [178], and 76.1 years [179], while the positive study of Duan et al. (2018) [177] used younger patients for a mean age of 73.2. As the risk of cognitive decline increases dramatically with age [180,181] or genetic variant status (APOE e4) [182], the amount of baseline neurodegeneration may have been substantially different in the two studies to mask any benefit. The study of Tae et al. (2019) [179] acknowledged that age was the strongest variable in the risk of dementia in all their comparisons. Another variable is the length of follow up. The positive study followed patients for 20 months [177], while the other two negative studies followed patients for 56 months [179] and 36 months [178]. Again, the two negative studies would have increased dementia at study end given the advanced age of the patients.

The amygdala can regulate psychological stressors and anxiety, besides regulating fear-conditioned memory and memory consolidation [7,183], and is regulated by the α_1A_-AR subtype [89,184]. Transgenic mice overexpressing the α_1A_-AR but not the α_1B_-AR showed antidepressant behavior [185]. α_1A_-AR blockage with WB4101 induces learned despair in mice [186] and tamsulosin facilitated depressive-like behavior in mice [187]. While a small clinical study found that tamsulosin decreased patient-reported depressive symptoms in BPH patients, contrary to the hypothesized effect in mice [188], BPH itself is associated with increased depressive and anxiety symptoms [189,190] and suicide [191]. Further large-scale clinical studies are needed to determine if tamsulosin and other α_1A_-AR blockers may increase depressive and anxiety-based disorders as hypothesized.

### 7.2. α_1_-AR Blockers May Increase Risk of Heart Failure

The Antihypertensive and Lipid-Lowering Treatment to Prevent Heart Attack Trial (ALLHAT) is a large, randomized double-blind study comparing four different classes of antihypertensive agents in patients older than 55 years [101]. The use of doxazosin (i.e., Cardura) increased the risk of stroke and the development of heart failure twice as much as those receiving a thiazide diuretic and caused this arm of the study to terminate early. In addition, doxazosin is not recommended as a first-line antihypertensive, particularly in the elderly [101,103]. However, this effect is not just isolated to doxazosin. A recent study of 175,200 men with BPH treated with either 5-alpha reductase inhibitors, various α_1_-AR antagonists, or a combination, found a 22% increased risk of cardiac failure among the users of α_1_-AR blockers [192]. Non-selective α_1_-AR blockers (terazosin, doxazosin, and alfuzosin) were significantly associated with an 8% higher risk for heart failure compared with selective α_1A_-AR blockers (silodosin and tamsulosin). Silodosin is 500-fold more selective for the α_1A_-AR than α_1B_-AR [193], while tamsulosin is 10-fold selective [145]. The α_1A_-AR is theorized to be cardioprotective and agonists protect against heart failure [56], but why are α_1A_-AR blockers then not associated with a higher risk of heart failure compared to non-selective blockers? There may be other non-α_1_-AR mediated effects associated with the increased risk of heart failure, such as increased apoptosis [148,149,150], particularly with α_1_-AR quinazoline blockers. While α_1_-AR blockers are still a popular treatment for BPH, and particularly in younger men who may not display heart failure, it is advised that physicians assess the cardiovascular health of the patient before long-term use.

### 7.3. α_1A_-AR Blockers May Have Adverse Ocular Effects

Another adverse effect of the long-term use of α_1_-AR antagonists is intraoperative floppy iris syndrome (IFIS), that increases serious complications and is characterized by a poor pupillary response, iris billowing, and prolapse during cataract surgery [194]. α_1_-ARs, and particularly the α_1A_-AR subtype, regulates the dilator smooth muscle of the iris [195,196], intraocular pressure [197,198], and the extracellular matrix and metabolic functions in human retinal pigment epithelium cells [199]. Tamsulosin has been identified to causing IFIS among BPH patients, with risks increased up to forty times more compared to other α_1_-AR antagonists and causing severe IFIS [200,201,202,203,204], but other non-selective α_1_-AR antagonists can also cause it. A large meta-analysis of over 6000 cases using various α_1_-AR antagonists indicate that most α_1_-AR blockers associate with a higher risk of IFIS [205]. With the increasing prevalence of both BPH and cataracts in the aging population, it is recommended that tamsulosin use is stopped 2 weeks before cataract surgery or is replaced by another α_1_-AR blocker.

## 8. Summary

The use of α_1_-AR agonists to potentially treat heart failure, cardiac ischemia, Alzheimer’s disease, and other dementias are targeted to the α_1__A_-AR subtype. However, all of these studies are preclinical in cell lines and mouse models or in initial clinical trials and it is not currently recommended to use these agents for non-approved use. Current development of positive allosteric modulators would be the choice as first-in-class therapeutics to avoid issues with increasing blood pressure to reduce other adverse side effects. The use of non-selective α_1_-AR antagonists of the quinazoline class to treat severe COVID-19/SARS, PTSD, and neurodegenerative disorders, such as Parkinson’s disease and ALS, have extensive evidence of efficacy in many clinical trials. However, the mechanism of action may be non-α_1_-AR mediated. Counterindications for α_1_-AR blockers are focused on those with established heart disease. Future clinical studies and larger, randomized, cross-over trials are required before drawing firmer conclusions about the counterindications of tamsulosin or other α_1A_-AR selective blockers.

## Figures and Tables

**Table 1 ijms-24-04188-t001:** α_1_-Adrenergic Receptor Agonists and Antagonists.

Drug	Receptor Selectivity	Current Indications	Potential Indications
Agonists			
**Non-selective**			
Norepinephrine	α_1_ = α_2_ = β	Septic and refractory	
Epinephrine	α_1_ = α_2_ = β	shock, Cardiopulmonary arrest	
		Hypotension	
**Selective**			
Phenylephrine	α_1_ > α_2_ >> β	Pupil dilation, Rosacea	
Oxymetazoline	α_1A_ > α_1D_ = α_1B_	Nasal decongestion, Rosacea	
Methoxamine	α_1A_ > α_1D_ > α_1B_	Septic and refractory shock	
Cirazoline	α_1A_ > α_1D_ > α_1B_		HF, Ischemia, cataracts
A-61603	α_1A_ > α_1D_ = α_1B_		HF, Ischemia, cataracts
Dabuzalgron	α_1A_ >> α_1D_ = α_1B_		HF, Ischemia, cataracts
Cmpd-3 ^1^	α_1A_ >> α_1D_ > α_1B_		AD, HF, Ischemia, cataracts
Antagonists			
**Non-selective**			
Prazosin	α_1A_ = α_1D_ = α_1B_	BPH, Therapy-resistant	COVID-19/SARS, PD,
Doxazosin	α_1A_ = α_1D_ = α_1B_	Hypertension,	ALS, PTSD,
Terazosin	α_1A_ = α_1D_ = α_1B_	Pheochromocytoma	Hyperinflammation
Alfuzosin	α_1A_ = α_1D_ = α_1B_		
**Selective**			
BMY7378	α_1D_ > α_1A_ >> α_1B_		
Tamsulosin	α_1A_ = α_1D_ > α_1B_	BPH, Pheochromocytoma	
Silodosin	α_1A_ > α_1D_ >> α_1B_	BPH	
5-Methylurapidil	α_1A_ > α_1D_ > α_1B_		
WB4101	α_1A_ = α_1D_ > α_1B_		

^1^ [12]. AD, Alzheimer’s disease; ALS, amyotrophic lateral sclerosis; BPH, benign prostatic hyperplasia; HF, heart failure; PD, Parkinson’s disease; PTSD, posttraumatic stress disorder; SARS, severe acute syndrome coronavirus 2.

## Data Availability

Not applicable.

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
