# Peer review of "α_1_-Adrenergic Receptors: Insights into Potential Therapeutic Opportunities for COVID-19, Heart Failure, and Alzheimer’s Disease"

_ijms, 2023, doi:10.3390/ijms24044188_

Round 1

Reviewer 1 Report

The manuscript is interesting and generally well written but it presents some flaws that must be resolved. In particular:

Table 1: table size must be reduced

Line 194:  "other organs " is not a good definition of the complexity of COVID-19 disease. It deserves to be highlighted that SARS-CoV-2 virus can lead to respiratory but also non-respiratory disease as recently review (PMID: 35943095, 35114008, 36072173).This is an important point to add since this is a review article. 

A summary table summarizing the studies on the α1A-AR agonists in each pathology discussed would be useful

Reference must follow the journal style

A bit of dedication in filling the IJMS template would have been appreciated since the author did not even deleted/modified the Author Contributions, Funding and Acknowledgments sections

Author Response

Table 1: table size must be reduced

I cannot reduce the table in the formatted text I received back from the journal. I believe that the table is large on purpose for review and then will be reformated according to their specifications after acceptance.

Line 194:  "other organs " is not a good definition of the complexity of COVID-19 disease. It deserves to be highlighted that SARS-CoV-2 virus can lead to respiratory but also non-respiratory disease as recently review (PMID: 35943095, 35114008, 36072173).This is an important point to add since this is a review article. 

We have modified this section and added more info on the multiple organ failures and included a new  review reference.

A summary table summarizing the studies on the α1A-AR agonists in each pathology discussed would be useful

Table 1 already includes a summary of pathology under “potential Indications “. Cirazoline, A-61603, Dabuzalgron, Cmpd-3, are all alpha1A-AR selective agonists and potential therapeutics listed are heart failure, Alzheimer’s Disease, Ischemia, and cataracts.

Reference must follow the journal style

We have now reformatted all of the references to journal style.

A bit of dedication in filling the IJMS template would have been appreciated since the author did not even deleted/modified the Author Contributions, Funding and Acknowledgments sections

That must have carried over from a previous publication of ours in this journal. Instructions stated that this section was to be filled for multiple authors.  As I am the sole author, I did not filled this out, but have now completed this.

Reviewer 2 Report

The manuscript's topic is interesting, and the review is timely and well-written. The author is a significant contributor to the alpha-1 adrenergic field, and I would like the author to consider a few aspects that likely were overlooked.

a) Some clinicians tend to be over-enthusiastic and could consider the review as a recommendation to use adrenergic agents to treat covid-19, heart failure, and some neurological diseases. The author clearly states that these are "potential treatments". Nevertheless, I suggest the author state that most information is from experiments with cultured cells and rodent models; human studies are either retrospective analyses or initial clinical studies with institutionally approved protocols. Emphasis should be made on these being promising potential therapeutic agents and that they are not presently recommended for treatment. Manuscript reading by a colleague with clinical activity could be helpful.

b) There are phrases I do not understand, and this could be due to errors/ typos that need to be corrected:

i)   Abstract:    ....activation by norepinephrine.....

ii) Table 1. BMY7378 is usually considered an antagonist (although it has some inverse agonist activity). Please consider or clarify.

iii) Section 3. Beta-ARs couple more efficiently to Gs...... (alpha is indicated)

iv) Section 5.1. .....dilate the iris...... (eye might also be correct, but the iris seems to be better)

v) Section 6.2.4 .... improve cellular pathophysiology in disorders.... seems confusing, ... "improve cellular function in disorders......" could be more precise.

vi) Section 6.4,  "alpha-1A-AR activation but alpha-1B-AR blockage is protective", is unclear to me.

vii) Section 7.1. "alpha-1A-AR but not non-selective blockers may increase dementia and depression"; "alpha-1A-AR blockers but not non-selective antagonists may increase dementia and depression" could be more precise.

viii) Section 7.2 "was a large" or "is a large"

ix) The author contributions, funding, and acknowledgments sections should be corrected.

Author Response

  1. a) Some clinicians tend to be over-enthusiastic and could consider the review as a recommendation to use adrenergic agents to treat covid-19, heart failure, and some neurological diseases. The author clearly states that these are "potential treatments". Nevertheless, I suggest the author state that most information is from experiments with cultured cells and rodent models; human studies are either retrospective analyses or initial clinical studies with institutionally approved protocols. Emphasis should be made on these being promising potential therapeutic agents and that they are not presently recommended for treatment. Manuscript reading by a colleague with clinical activity could be helpful.

We have added the following lines to both the abstract and summary. “While the studies reviewed here are still preclinical in cell lines and rodent disease models or have undergone initial clinical trials, potential therapeutics discussed here should not be used for non-approved conditions.”

  1. b) There are phrases I do not understand, and this could be due to errors/ typos that need to be corrected:
  2. i)   Abstract:    ....activation by....

corrected

  1. ii) Table 1. BMY7378 is usually considered an antagonist (although it has some inverse agonist activity). Please consider or clarify.

It is indeed an antagonist, this is now corrected in Table 1

iii) Section 3. Beta-ARs couple more efficiently to Gs...... (alpha is indicated)

corrected

  1. iv) Section 5.1. .....dilate the iris...... (eye might also be correct, but the iris seems to be better)

corrected

  1. v) Section 6.2.4 .... improve cellular pathophysiology in disorders.... seems confusing, ... "improve cellular function in disorders......" could be more precise.

corrected

  1. vi) Section 6.4,  "alpha-1A-AR activation but alpha-1B-AR blockage is protective", is unclear to me.

We have now inserted this statement:  Concerning α1-AR subtype-dependent effects of antagonists, there is evidence that α1A-AR activation is protective while chronic α1B-AR activation is damaging and neurodegenerative.  Therefore,  α1A-AR agonists would be protective and systems where chronic α1B-AR activation is damaging, non-selective blockers may exert protective effects”

vii) Section 7.1. "alpha-1A-AR but not non-selective blockers may increase dementia and depression"; "alpha-1A-AR blockers but not non-selective antagonists may increase dementia and depression" could be more precise.

corrected

viii) Section 7.2 "was a large" or "is a large"             

corrected

  1. ix) The author contributions, funding, and acknowledgments sections should be corrected.

That must have carried over from a previous publication in this journal. Instructions stated that this section was to be filled for multiple authors.  As I am the sole author, I did not filled this out, but have now completed this.

Round 2

Reviewer 1 Report

the manuscript has been significantly improved and can be accepted in the present form